# Incidence, Hospitalization, Mortality and Risk Factors of COVID-19 in Long-Term Care Residential Homes for Patients with Chronic Mental Illness

Alberto Arnedo-Pena [1,2,3,*] , María Angeles Romeu-Garcia [1], Juan Carlos Gasco-Laborda [1],
Noemi Meseguer-Ferrer [1], Lourdes Safont-Adsuara [1], Francisco Guillen-Grima [3] ,
María Dolores Tirado-Balaguer [4], Susana Sabater-Vidal [4], María Gil-Fortuño [5], Oscar Pérez-Olaso [5],
Noelia Hernández-Pérez [5], Rosario Moreno-Muñoz [4] and Juan Bellido-Blasco [1,2,6]

1    Epidemiology Division, Public Health Center, 12003 Castelló de la Plana, Spain
2    Public Health and Epidemiology (CIBERESP), 28029 Madrid, Spain
3    Department of Health Sciences, Public University of Navarra, 31006 Pamplona, Spain
4    Microbiology Laboratory, Universitary General Hospital, 12004 Castelló de la Plana, Spain
5    Microbiology Laboratory, Universitary Hospital de la Plana, 12540 Vila-Real, Spain
6    Department of Epidemiology, School of Medicine, Jaume I University, 12006 Castelló de la Plana, Spain
*    Correspondence: albertoarnedopena@gmail.com

**Abstract:** Long-term care residential homes (LTCRH) for patients with chronic mental illness have suffered the enormous impact of COVID-19. This study aimed to estimate incidence, hospitalization, mortality, and risk factors of COVID-19 to prevent future epidemics. From March 2020 to January 2021 and before vaccination anti-SARS-CoV-2 begins, cumulate incidence rate (CIR), hospitalization rate (HR), mortality rate (MR), and risk factors of COVID-19 in the 11 LTCRH of two Health Departments of Castellon (Spain) were studied by epidemiological surveillance and an ecological design. Laboratory tests confirmed COVID-19 cases, and multilevel Poisson regression models were employed. All LTCRH participated and comprised 346 residents and 482 staff. Residents had a mean age of 47 years, 40% women, and suffered 75 cases of COVID-19 (CIR = 21.7%), five hospitalizations (HR = 1.4%), and two deaths (MR = 0.6%) with 2.5% fatality-case. Staff suffered 74 cases of the disease (CIR = 15.4%), one hospitalization (HR = 0.2%), and no deaths were reported. Risk factors associated with COVID-19 incidence in residents were private ownership, severe disability, residents be younger, CIR in municipalities where LTCRH was located, CIR in staff, and older age of the facilities. Conclusion: COVID-19 incidence could be prevented by improving infection control in residents and staff and modernizing facilities with increased public ownership.

**Keywords:** COVID-19; long-term care residential homes; mental illness; intellectual disability; incidence hospitalization; mortality; risk factors; epidemiological surveillance; ecologic design

## 1. Introduction

Along with nursing homes (NH), long-term care residential homes (LTCRH) for patients with chronic mental illness have suffered the enormous impact of the COVID-19 pandemic. The mortality was generally lower than in the NH, but the morbidity was elevated [1–4]. In general, the LTCRH were not prepared to affront an epidemic such as COVID-19. The SARS-CoV-2 presents an elevated transmission by exposure to respiratory droplets and aerosol particles that contain the virus and contact with contaminated surfaces and fomites [5]. A case could be infective 2 or 3 days before symptoms; there are asymptomatic cases in a high proportion [6], and the virus could remain in a patient for a long period. However, some aspects of the epidemiology and transmission remain unknown, including the origin of the virus, the role of asymptomatic carriers, the duration of the immunity after infection, and the sources of virus variants [7].

Residents in LTCRH are regarded as COVID-19 high risk considering the prevalence of immunity deficiencies, and mental and physical disability with some difficulty to perform measures of prevention such as hygiene practices, wearing a mask, or physical distancing. Usually, the staffs in LTCRH were young with low specialization, and many residents need a high level of personal care assistance. In addition, infection control measures, including the use of personal protective equipment, were insufficient in the first wave. With respect to the facilities, the capacity for isolation, compartmentalization, and quarantines were difficult to implement because of the architecture of LTCRH, their age of the buildings, the high occupancy and their ownership that could explain the high transmission in LTCRH [8–11]. However, studies on the incidence and mortality of COVID-19 in these centers are scarce compared to NH's studies, and the risk factors are not well established [12–14].

From March 2020 to January 2021, before the vaccination against SARS-COV-2, the LTCRH in the Health Departments (HD) 2 and 3 of Castellon (Spain) were significantly affected by the COVID-19 pandemic. These centers care for patients from 18 to 65 years old who suffer chronic mental illness, including intellectual and developmental disability, autism spectrum disorder, Down syndrome, and cerebral palsy, but they do not require hospitalization [15].

This study aimed to estimate the incidence, hospitalizations, mortality, and risk factors of the COVID-19 pandemic in the LTCRH in the first year of the COVID-19 pandemic in order to establish preventive measures to future epidemics.

## 2. Materials and Methods

The study presents two parts. In the first part, by the epidemiologic surveillance of COVID-19, the HD of Castellon reported incident COVID-19 cases, hospitalizations, and mortality during the study period March 2020–January 2021. In the second part, aggregate data of LTCRH were obtained, including resident's characteristics, staff, and facilities, to perform an ecologic design with each LTCRH as an analysis unit. The study included all LTCRH in the HD Castellon, less a LTCRH center, which was located at the same institution that a NH, and this center was incorporated into the study of COVID-19 in the NH in Castellon [16].The study followed the STROBE guidelines of observational epidemiologic studies [17].

The study was performed by the Epidemiology Division and social workers of the Public Health Center of Castellon. Reported cases, hospitalizations, and deaths due to COVID-19 were studied, and COVID-19 outbreaks of LTCRH were detected.

An outbreak was reported to the Epidemiology Division when a COVID-19 case in a LTCRH. Commercial molecular tests based on real time reverse transcriptase-polymerase chain reaction (RT-PCR) Roche Lightmix Modular SARS-CoV-2 (Roche-TIB MOLBIOL D-12103 Berlin Germany), VIASURE SARS-CoV-2 Real Time PCR Detection Kit (CerTest Biotec S.L, San Mateo de Gallego, Zaragoza, Spain), Abbott Real-Time SARS-CoV-2 (Abbott Laboratory, Abbott Park, IL, USA), and Argene SARS-CoV-2 R-Gen (Biomérieux, F-69280, Marcy l'Etoile, France), rapid antigen detection assays (PANBIO™ COVID-19 Ag, Abbott Laboratory, Abbott Park, IL, USA) and chemiluminescent microparticle immunoassay (CMIA) for IgG-antibodies(anti-S and/or anti-N), and/or IgM-antibodies (anti-S) detection (Alinity, Abbott Laboratory, Abbott Park, IL, USA) were used to confirm COVID-19 cases [18–20]. The analysis tests were done at the Microbiology Service of University General Hospital in Castellon and a lower proportion at the Microbiology Section of University Hospital de la Plana in Vila-real.

In order to obtain the LTCRH characteristics of residents and staff, the social workers carried out a specific questionnaire, and the LTCRH directions completed it. In this questionnaire, different variables were collected: numbers of residents, mean age, numbers of women and men, disability grade, number of staff and occupation distribution, age of the building, numbers of beds, numbers of bedrooms (singles, doubles, three beds), numbers of bathrooms, total of bathrooms, the capacity to compartmentalize with zones, COVID-19

patients' isolation, and quarantines in the LTCRH. In addition, contingency plans prepared by each LTCRH for preventing the COVID-19 pandemic were consulted.

The median, ranges, mean, and standard deviation were used to describe the variables. The cumulative incidence rates (CIR) of COVID-19 in residents and staff were calculated considering the total number of reported cases divided by residents or staff who were residing or working in each LTCRH when the first case of COVID-19 was reported. The hospitalization rate (HR) was calculated by dividing the number of hospitalized cases by the number of residents or staff. The mortality rate (MR) of COVID-19 was calculated by dividing the total number of deaths by the total number of residents residing at the LTCRH in the first outbreak. The case fatality was calculated by dividing the number of deaths from COVID-19 by the number of patients from COVID-19. During the study period, new entries to the LTCRH, especially of residents, have been considered very few due to the pandemic outbreak. Then, the indicated rates in some centers could increase due to maintaining the initial population. CIR, HR, and MR were the dependent variables, and the independent variables were the following:

- Residents: mean age (years), women percentage (%), degree three of disability (%).The used index of disability, based on level of dependence, comprehended four degrees: 0 (no disability); 1 minor disability (mild dependence); 2 moderate disabilities (maintains autonomy in daily life); 3 severe/profound disabilities. Degree three of disability needs help for functions such as grooming, dressing, feeding, or going to bed.
- Staff: number of total staff, registered nurses, and nursing assistants. Ratios: residents/staff, registered nurses/residents, nursing assistant/residents, staff/beds, nurse assistant /beds, and residents/bathrooms.
- LTCRH: ownership (private or public), size (number of residents less than 30, or $\geq$30), the municipality where the LTCRH was located, large urban area ($\geq$50,000 inhabitants), urban area ($\geq$10,000–49,999 inhabitants), semi-urban area ($\geq$2500–9999 inhabitants), and rural area ($\leq$2500 inhabitants). LTCRH facilities: number of beds, baths, and bathrooms; in addition, number of single and double bedrooms, and bedrooms with three beds. Occupancy rate: resident population/beds.

COVID-19 cumulative incidence rate per 100,000 inhabitants in each municipality for 2020 was collected. We used an adaptation of the crowding index considering the conditions LTCRH by the formula [21]:

Crowding index = number of residents in the first COVID-19 outbreak/[(number of bedrooms/2) + (number of bathrooms/2)].

The crowding index is a mean of singles, doubles, and three beds in bedrooms and bathrooms with the number of residents for each LTCRH. If the index is equal to or larger than 2.00, it could indicate crowding.

In the statistical analyses, we used Poisson regression models for the univariate analysis and multilevel Poisson regression models for the multivariate analysis, considering the location area with four levels as the reference of each LTCRH. Calculating crude and adjusted relative risks (cRR and aRR) with a 95% confidence interval (CI) as a measure of the associations between risk factors and the cumulative incidence was performed. The direct acyclic graph (DAG) approach with DAGitty version 3.0 (Johannes Textor, Nijmegen, The Netherlands) [22] was used to determine potential confounding factors in the multivariate analysis, and each independent variable was adjusted for the confounding factors following the DAGs. All statistical analysis was performed with Stata® version 14.2 (Stata Corp, College Station, TX, USA).

No Ethical Committee approval was necessary considering the epidemiological surveillance of the LTCRH during the COVID-19 pandemic period, the anonymity of all the participants, and the administrative information about each Center.

### 3. Results

In the study, all the 11 LTCRH of the Health Department 2 and 3 of Castellon participated. These centers were located in seven municipalities: Castellon de la Plana with four centers, Vila-real with two centers, and a center in Benicassim, Vall D'Uixo, Borriol, Albocasser, and Vilafames. Characteristics of residents, staff, and facilities of LTCRH are shown in Table 1. The age mean of residents was 47 years of the median (range 38–54), with more men than women (60%). Degree of disability were zero (3.6%), one (6.7%), two (25.0%), and three (55.6%).

**Table 1.** Characteristics of long-term care residential homes (LTCRH) for patients with chronic mental illness (II). Health Departments 2 and 3 of Castellon. March 2020–January 2021.

| Variables | |
|---|---|
| **Residents** | ***N* = 346** |
| Age median of means (range) | 47 years (38–54) |
| Women (%) | 40% (14–67%) |
| Disability | |
| 0 None | 3.6% (0–52.5%) |
| 1 Mild/Minor | 6.7% (0–54%) |
| 2 Moderate | 25.0% (0–46%) |
| 3 Severe/Profound | 55.6% (0–100%) |
| **Staff** | ***N* = 482** |
| Registered nurses | 23 (4.8%) |
| Nursing assistant | 202 (41.9%) |
| Centers for chronic mental illness | |
| Age of the building (years) | median 17 years (range 3–43 years) |
| Occupancy rate (median range) | 95.2% (52–100%) |
| Size of residents' places | median 28 (range 20–50) |
| Ratios (median and range) | |
| Residents/staff | 0.69 (0.47–1.11) |
| Registered nurses/residents | 0.09 (0–0.14) |
| Nursing assistant/residents | 0.57 (0–1.35) |
| Staff/beds | 1.25 (0.77–2.14) |
| Nursing assistant/beds | 0.50 (0–1.24) |
| Residents/bedrooms | 5.25 (0.85–10.0) |
| Crowding index | 2.65 (0.92–4.0) |
| LTCRH's capacity | |
| Compartmentalization | Yes: 6 (54.6%) No: 5 (45.4%) |
| Isolation | Yes: 9 (81.2%) No: 2 (18.2%) |
| Quarantines | Yes: 9 (81.8%) No: 2 (18.2%). |

In total, 346 residents live in the LTCRH with 482 staff, composed of 22 registered nursing (4.8%) and 202 nurse assistants (41.9%). The ownership was private in seven and public in four. The size of LTCRH had a median of 28 places of residents (range 20–50) for each center, with fewer than 30 places in six centers, from 30 to 39 in two centers, and more than 40 places in three centers. The age of LTCRH buildings had a median of 17 years (range 3–43 years). There were 226 bedrooms with 102 bedrooms with bathrooms (44.9%).

The bedrooms were 90 singles (23.4%), 113 doubles (58.7%), and 23 with three beds (17.9%) with a total of 385 beds.

Ratios of residents/staff had important differences among centers with a median of 0.69 (range 0.47–1.11) and ratio nurse assistant /residents median of 0.67 (range 0–1.24). The crowding index had a median of 2.65 (range 0.94–4.0), with a higher occupancy rate. The compartmentalization was only done in six LTCRH (54.5%). Isolation and quarantines would be made in nine centers (81.2%).

During the study period, 18 outbreaks of COVID-19 were reported in the LTCRH by the epidemiologic surveillance, considering that only a case of COVID-19 needed particular actions of control and prevention of the disease. In two LTCRH, no cases were reported in residents or staff, in three LTCRH only an outbreak, in three LTCRH two outbreaks, and in three LTCRH three outbreaks. The median cases by outbreaks were two (range 0–27) for residents and two (range 0–18) for staff. The five residents hospitalized occurred in three LTCRH with attack rates of outbreaks of 42.1%, 34.4%, and 55%, respectively. Laboratory confirmation of cases included 50 PCR-test, 22 SARS-CoV-2 antigen rapid tests, 3 SARS-CoV-2 antibodies for residents, and 58 PCR-test, 13 SARS-CoV-2 antigens rapid, and 3 SARS-CoV-2 antibodies for staff.

COVID-19 incidences, hospitalizations, and deaths are presented in Table 2. Seventy-five were reported with a CIR of 21.7% (95% CI 17.3–27.2%) for residents, and 74 cases with a CIR of 15.4% (95% IC 12.4–19.5%) for staff. The incidence of COVID-19 presented high variations from 0% CIR to 57.1% in residents and from 0% to 28.9% in staff. Among cases, five needed to be hospitalized (6.7%) in residents with a HR of 1.4% (95% CI 0.4–2.5%), and one hospitalization (1.4%) in staff with a HR of 0.2% (95% CI 0.01–1.50%). Two deaths occurred in residents, MR 0.6% (95% CI 0.1–2.3%) with a fatality-case of 2.5%, and were two males, 44 and 55 years old, affected by Down syndrome and mental retard, respectively.

**Table 2.** Incidence, hospitalization, and mortality COVID-19 in long-term care residential homes for patients with chronic mental illness (I). Health Departments 2 and 3 of Castellon. March 2020–January 2021.

| Total Population | COVID-19 Cases | Cumulative Incidence Rate 95% Confidence Interval |
|---|---|---|
| Residents *n* = 346 | 75 | 21.7% (17.3–27.2%) |
| Staff *n* = 482 | 74 | 15.4% (12.4–19.5%) |
| | Hospitalizations | Hospitalization rate |
| Residents | 5 | 1.4% (0.4–2.5%) |
| Staff | 1 | 0.2% (0.01–1.50%) |
| | Deaths | Mortality rate |
| Residents | 2 | 0.6% (0.1–2.3%) |
| Staff | 0 | 0.0 |

Risk factors of COVID-19 in the residents of LTCRH are shown in Tables 3 and 4 by crude and adjusted RR. Considering adjusted RR, CIR was higher in private than public LTCRH (aRR = 40.6 95% CI 2.87–574.08), in LTCRH with high percentage of residents with degree three of disability (aRR = 3.15 (95% CI 1.25–7.93), and age mean of residents was younger (aRR = 0.81 95% CI 0.76–0.87). The LTCRH size, urban localization, or the percentage of resident women was not associated with CIR.

**Table 3.** Risk factor of COVID-19 incidence in long-term care residential homes for patients with chronic mental illness (I). Health Departments 2 and 3 of Castellon. March 2020–January 2021.

| | CIR [1] | c [2] RR [3] 95% IC [4] | *p*-Value | a [5] RR 95% IC | *p*-Value |
|---|---|---|---|---|---|
| Ownership | | | | | |
| Private [6] | **25.8%** | **1.25 (0.79–2.00)** | **0.342** | **40.6(2.87–574.08)** | **0.006** |
| Public | 18.9% | 1.00 | | 1.00 | |
| Size [7] | | | | | |
| >28 residents | 22.8% | 0.91 (0.58–1.43) | 0.676 | 3.93 (0.01–2345.29) | 0.675 |
| <28 residents | 22.9% | 1.00 | | 1.00 | |
| Large urban area [8] | | | | | |
| ≥50,000 inhabitants. | 17.1% | 0.51 (0.32–0.81) | 0.004 | 0.61 (0.11–3.52) | 0.582 |
| <50,000 inhabitants. | 30.0% | 1.00 | | 1.00 | |
| Residents | | | | | |
| SevereDisability [9,10] > 55.6% | **40.5%** [13] | **4. 93 (2.80–8.68)** | **0.000** | **3.15 (1.25–7.93)** | **0.015** |
| ≤55.6% | 1.1% [13] | 1.00 | | 1.00 | |
| Age (median) [10,11] > 47 years | **9.5%** [13] | **0.87 (0.82–0.92)** | **0.000** | **0.81 (0.76–0.87)** | **0.000** |
| ≤47 years | 34.4% [13] | 1.00 | | 1.00 | |
| % Women [10,12] > 30% | 11.8% [13] | 0.29 (0.04–2.08) | 0.220 | 17.2 (0.98–303.21.48) | 0.052 |
| ≤30% | 34.4% [13] | | | | |

[1] CIR = cumulative incidence rate (Poisson regression) [2] c = crude. [3] RR = relative risk. [4] CI = Confidence interval. [5] Adjusted. [6] Adjusted for mean age % women disability. [7] Adjusted for mean age % women disability ownership. [8] Adjusted for mean age % women. [9] Adjusted for mean age % women. [10] Median of the variable. [11] Adjusted for % women. [12] Adjusted for mean age. [13] Cumulative incidence rates.

**Table 4.** Risk factor of COVID-19 incidence in long-term care residential homes (LTCRH) for patients with chronic mental illness. Health Departments 2 and 3 of Castellon. March 2020–January 2021.

| | CIR [1] | c [2] RR [3] 95% CI [4] | *p*-Value | a [5] RR 95% CI | *p*-Value |
|---|---|---|---|---|---|
| CIR municipalities [6,7] > 2944·10[5] | **27.9%** | **1.05 (1.07–1.20)** | **0.004** | **1.34 (1.16–1.55)** | **0.000** |
| ≤2944·10[5] | 16.7% | 1.00 | | 1.00 | |
| CIR staff [6,8] mean > 10% | **46.7%** | **7.87 (4.15–14.93)** | **0.000** | **6.61 (2.64–16.54)** | **0.000** |
| ≤10% | 1.1% | 1.00 | | 1.00 | |
| Age building [6,9] > 17 years | **42.1%** | **1.04 (1.02–1.07)** | **0.000** | **1.10 (1.05–1.15)** | **0.000** |
| ≤17 years | 19.1% | 1.00 | | 1.00 | |
| Occupancy rate % [6,9] > 95.2% | 1.1% | 0.79 (0.17–3.60) | 0.758 | 0.61 (0.01–62.66) | 0.833 |
| ≤95.2% | 42.1% | 1.00 | | 1.00 | |
| Residents/staff [6,9] > 0.634 | 12.7% | 0.05 (0.01–0.22) | 0.000 | 1.26 (0.00–57,254.0) | 0.967 |
| ≤0.634 | 35.0% | 1.00 | | 1.00 | |
| Nursing assistant/residents ratio [6,9] > 0.57 | 34.4% | 3.50 (1.91–6.43) | 0.000 | 0.32 (0.01–26,23) | 0.618 |
| ≤0.57 | 10.7% | 1.00 | | 1.00 | |
| Staff/beds ratio [6,9] > 1.25 | **46.7%** | **3.67 (2.12–6.37)** | **0.000** | **1.02 (0.16–6.48)** | **0.981** |
| ≤1.25 | 8.35% | 1.00 | | 1.00 | |

| | CIR [1] | c [2] RR [3] 95% CI [4] | *p*-Value | a [5] RR 95% CI | *p*-Value |
|---|---|---|---|---|---|
| Nursing assistants/beds ratio [6,9] > 0.5 | 42.1% | 3.95 (2.11–7.39) | 0.000 | 0.62 (0.04–8.51) | 0.718 |
| ≤0.5 | 0.0% | 1.00 | | 1.00 | |
| Residents/total bathrooms ratio [6,9] > 5.25 | 42.1% | 1.17 (1.10–1.26) | 0.000 | 1.17 (0.84–1.63) | 0.352 |
| ≤5.25 | 3.9% | 1.00 | | 1.00 | |
| Crowding index [6,9] > 2.65 | 42.1% | 1.46 (1.15–1.85) | 0.002 | 1.48 (0.32–6.77) | 0.612 |
| ≤2.65 | 1.1% | 1.00 | | 1.00 | |

[1] CIR = Cumulative incidence rate (median). [2] c = crude. [3] RR = relative risk [4] CI = Confidence interval. [5] Adjusted. [6] Median of the variable. [7] Adjusted for mean age % women disability LTCRH size ownership. [8] Adjusted for mean age % women disability. [9] Adjusted for mean age % women disability ownership LTCRH size.

In Table 4, other risk factors of CIR in residents are shown. The crude analysis found a high number of significant associations between risk factors and CIR. However, when an adjusted analysis was performed, only three factors were associated with an increase in COVID-19: the CIR of the municipality where the LTCRH was located (aRR = 1.34 95% CI 1.16–1.55), the CIR of staff (aRR = 6.61 95% CI 2.64–16.54), and higher age of the LTCRH building (aRR =1.10 95% CI 1.05–115).

The occupancy rate, ratios of residents/staff, nursing assistants, ratios of staff or nursing assistants/beds, the ratio of residents/bathrooms, and the crowding index were not associated with COVID-19.

The small numbers of hospitalizations and death prevent performing an analysis of risk factors, considering that the results are volatile and do not allow valid estimates.

## 4. Discussion

The results of this study indicate that COVID-19 incidence in LTCRH was high but less than the reported incidence in nursing homes of Castellon, 21.7% versus 34.8% of CIR. However, COVID-19 mortality rate was much higher in nursing homes than in LTCRH in Castellon, 8.7% versus 0.6% [16]. The means of ratios for residents/staff and nursing assistants/residents were agreed to the legislation of Valencia Generalitat, and eight LTCRH have less than 40 residents, the maximum size allowed for LTCRH [15]. Although the ratio of registered nurses/residents was low, residents/staff ratios were better than Castellon's NH [16].

The elevated COVID-19 incidence in residents is consistent with other studies of patients with mental illness with respect to incidence [23–25], hospitalization [26–28], and mortality [29,30], and it is higher than reported in the general population of Castellon during the study period, 2.9%, that supposes a RR of 7.5. In addition, the hospitalization and mortality rates of LTCRH were 1.4% and 0.6%, respectively, versus 0.3% and 0.06% in the general population of Castellon; the RR were 4.7 and 10.0, respectively.

In the study, risk factors of COVID-19 incidence may establish three groups: the characteristic of residents such as severe disability and young age, COVID-19 incidence in the community and the staff, and conditions of the facility, age of the building, and private ownership. Examining the strength of the association, private ownership, COVID-19 incidence in the staff and severe disability in the residents were the more important risk factors. Private ownership as a risk factor is indicated in studies of long-term care homes considering the high occupancy, the inadequate facilities, and regarding older design standards than the public ownership centers [31,32]. The COVID-19 incidence in the staff have been found to be associated with COVID-19 outbreaks in these centers [33,34]. The severe disability was found in nursing homes, and it is well known risk factor [35–37]. These factors could be related to the impossibility of compartmentalization in five LTCRH by architectonic design, and two LTCRH were unable to isolate COVID-19 patients and quarantines.

Other risk factors found in our study were a high incidence of COVID-19 in municipalities where LTCRH were located and the high age of facility buildings. These findings are in line with the results of centers for people with psychological disabilities [38], long-term care facilities [39,40], and considering the aging of the facilities [41]. An older age of residents was a protection factor in line with the studies in United States [42] and Canada [43].

Concerning risk factors of COVID-19 incidence, important differences with risk factors of NHs are highlighted in the ratios of residents/staff, nurse assistants, and nurse assistants/beds, crowding index, and size [16,21,44,45]. All these factors were not associated with COVID-19 in this study. However, the large size of these LTCRH has been associated with an increase in COVID-19 incidence [46,47]. In our study, the LTCRH's median of residents was low and could be a protector factor.

The study's limitations include: First, the estimation of the CIR considering the population of residents in the first COVID-19 outbreak could increase the CIR, but in the study period, the entrances and exits of residents were reduced by the pandemic situation. Second, we use aggregated data to estimate risk factors and not individual data. Third, no information on staff horary could be analyzed. Fourth, the studied risk factors of residents were limited, and the few hospitalizations and deaths prevented their statistical analyses to find associated factors. Finally, COVID-19 is a new disease, and other factors not considered here can be crucial in some aspects.

## 5. Conclusions

In conclusion, COVID-19 transmission in LTCRH was associated with staff and municipality infection, disability of residents, young residents, older facility age, and private ownership. In addition, the limitation of compartmentalization, isolation, and quarantines are related to the difficulty of adequately implementing control and preventing infections. The recommendations address the importance of adequate infection control (strict hygiene and early case detection, serial screening), improved architectonic design, increased LTCRH public ownership with more governmental implication in the care home sector [48], and the vaccination against SARS-CoV-2 for residents and staff as a priority [49–51].

**Author Contributions:** Conceptualization, A.A.-P. and J.B.-B.; methodology, A.A.-P., M.A.R.-G., J.C.G.-L. and J.B.-B., software, A.A.-P., J.C.G.-L., N.M.-F., L.S.-A. and N.H.-P.; validation, N.M.-F., L.S.-A. and M.D.T.-B., formal analysis, A.A.-P. and F.G-G.; investigation, M.A.R.-G., J.C.G.-L., M.D.T.-B., S.S.-V., M.G.-F., O.P.-O., N.H.-P. and J.B.-B.; resources, J.B.-B., R.M.-M. and M.G.-F.; data curation, N.M.-F., writing—original draft preparation, A.A.-P. and M.A.R.-G.; writing—review and editing, M.D.T.-B., S.S.-V., R.M.-M., M.G.-F., N.M.-F., F.G.-G. and J.B.-B.; visualization, N.M.-F., S.S.-V., F.G-G. and M.G.-F.; supervision, F.G.-G., J.B.-B. and R.M.-M.; project administration, N.M.-F. and L.S.-A.; funding acquisition, J.B.-B. and R.M.-M. All authors have read and agreed to the published version of the manuscript.

**Funding:** This research received no external funding.

**Institutional Review Board Statement:** Ethical review and approval were waived for this study due to the epidemiologic surveillance of the LTCRH motivated by the COVID-19 pandemic according to Spanish laws and regulations [52–55].

**Informed Consent Statement:** Patient consent was waived due to surveillance epidemiological of LTCRH motivated by the COVID-19 pandemic, and the authors used aggregated data of each LTCRH.

**Data Availability Statement:** Authorization of the Public Health Center's direction will be required to consult the data set of this study.

**Acknowledgments:** The authors thank the managers of LTCRH in the Health Departments 2 and 3 of Castellon for their help and support in the implementation of this study and express gratitude to the residents and staff of the 11 LTCRH. In addition, the authors appreciate the work of Laura Prades-Vila, Matilde Flores-Medina, and Viorica Rusen in carrying out this study.

**Conflicts of Interest:** The authors declare no conflict of interest.

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
