# Peer review of "Incidence, Hospitalization, Mortality and Risk Factors of COVID-19 in Long-Term Care Residential Homes for Patients with Chronic Mental Illness"

_epidemiologia, doi:10.3390/epidemiologia3030030_

Round 1

Reviewer 1 Report

The present manuscript describes epidemiological characteristics of COVID-19 in Long-term care residential homes for patients with chronic mental illness and aims to highlight risk factors of COVID-19 in these facilities in order to prevent upcoming outbreaks. This study had comprised 11 LTCRH of the Health Department 2 and 3 of Castellon and shown that COVID transmission in the LTCRH was linked to staff and municipality infection, disability of residents, young residents, older facility age, and private ownership. To my knowledge, this is the first study on COVID-19 transmission in LTCRH in Spain. The work is well conducted, with appropriated statistic tests. The results are convincing and seem trustable. Thus, the whole “Material and methods” and “results” section is perfectly correct.

However, the introduction section is too short and the scientific context of the research is not sufficiently presented.

In the discussion section results are not adequately discussed in relation to the relevant bibliography and literature citation is poor. There are several papers dealing with care homes and the risk of COVID-19 in Europe and around the world, which the authors seem not to be aware of, such as:

Daly, M. (2020). COVID‐19 and care homes in England: What happened and why?. Social Policy & Administration54(7), 985-998.

Stall, N. M., Jones, A., Brown, K. A., Rochon, P. A., & Costa, A. P. (2020). For-profit long-term care homes and the risk of COVID-19 outbreaks and resident deaths. Cmaj192(33), E946-E955.

Burton, J. K., Bayne, G., Evans, C., Garbe, F., Gorman, D., Honhold, N., ... & Guthrie, B. (2020). Evolution and effects of COVID-19 outbreaks in care homes: a population analysis in 189 care homes in one geographical region of the UK. The Lancet Healthy Longevity1(1), e21-e31.

Comas-Herrera, A., Zalakaín, J., Lemmon, E., Henderson, D., Litwin, C., Hsu, A. T., ... & Fernández, J. L. (2020). Mortality associated with COVID-19 in care homes: international evidence. Article in LTCcovid. org, international long-term care policy network, CPEC-LSE14.

Logar, S. (2020). Care home facilities as new COVID-19 hotspots: Lombardy Region (Italy) case study. Archives of gerontology and geriatrics89, 104087.

And others

MINOR POINTS

-Abstract:

Line 22:  “and risk factors of COVID-19 for prevent future epidemics”.  (“and risk factors of COVID-19 to prevent future epidemics”

line 27 " All LTCRH participanted and" (All LTCRH participated and);

introduction

line 45 “ However, the studies on incidence” ( However, studies on incidence)

line 62 and 63 “unit. All LTCRH in the Health Departments were studied, less a center located in an institution that included an NH and was included in our study of COVID-19 in NH” the statement requires clarification

Results

Replace Si by yes in the table 1

Line 152 “A with fewer than 30 places in 6 centers,” (With fewer than 30 places….)

Lines 219-221 “In addition, the means of ratios for residents/staff and nursing assis-219 tants/residents in the LTCRH were according to the legislation of Valencia Generali-220 tat[10]. However, the ratio of registered nurses/residents was low, and two centers had 221 more than 40 residents, the maximum size allowed for these LTCRH[10]”.

Would you clarify the relationship between this statement and the above mentioned results!

Line 236 “results of Gorgels y co-authors [31]” (results of Gorgels and co-authors [31])

Author Response

  1. First Reviewer.

The present manuscript describes epidemiological characteristics of COVID-19 in Long-term care residential homes for patients with chronic mental illness and aims to highlight risk factors of COVID-19 in these facilities in order to prevent upcoming outbreaks. This study had comprised 11 LTCRH of the Health Department 2 and 3 of Castellon and shown that COVID transmission in the LTCRH was linked to staff and municipality infection, disability of residents, young residents, older facility age, and private ownership. To my knowledge, this is the first study on COVID-19 transmission in LTCRH in Spain. The work is well conducted, with appropriated statistic tests. The results are convincing and seem trustable. Thus, the whole “Material and methods” and “results” section is perfectly correct.

Thank you very much for your revision and for your indications.

However, the introduction section is too short and the scientific context of the research is not sufficiently presented.

The introduction is increased explaining the characteristic of COVID-19 transmission in long-term care residential homes and some references have been included. In addition, a more detailed description of residents, staff, and facilities was performed.

In the discussion section results are not adequately discussed in relation to the relevant bibliography and literature citation is poor. There are several papers dealing with care homes and the risk of COVID-19 in Europe and around the world, which the authors seem not to be aware of, such as:

Daly, M. (2020). COVID‐19 and care homes in England: What happened and why?. Social Policy & Administration54(7), 985-998.

Stall, N. M., Jones, A., Brown, K. A., Rochon, P. A., & Costa, A. P. (2020). For-profit long-term care homes and the risk of COVID-19 outbreaks and resident deaths. Cmaj192(33), E946-E955.

Burton, J. K., Bayne, G., Evans, C., Garbe, F., Gorman, D., Honhold, N., ... & Guthrie, B. (2020). Evolution and effects of COVID-19 outbreaks in care homes: a population analysis in 189 care homes in one geographical region of the UK. The Lancet Healthy Longevity1(1), e21-e31.

Comas-Herrera, A., Zalakaín, J., Lemmon, E., Henderson, D., Litwin, C., Hsu, A. T., ... & Fernández, J. L. (2020). Mortality associated with COVID-19 in care homes: international evidence. Article in LTCcovid. org, international long-term care policy network, CPEC-LSE14.

Logar, S. (2020). Care home facilities as new COVID-19 hotspots: Lombardy Region (Italy) case study. Archives of gerontology and geriatrics89, 104087.

And others

Thank you very much for your indications. We add the suggested references and we carried out a more detailed comparison with relevant publications in the discussion. 

MINOR POINTS

-Abstract:

Line 22:  “and risk factors of COVID-19 for prevent future epidemics”.  (“and risk factors of COVID-19 to prevent future epidemics”

line 27 " All LTCRH participanted and" (All LTCRH participated and);

introduction

line 45 “ However, the studies on incidence” ( However, studies on incidence)

-We have corrected all the mistakes and typos.

line 62 and 63 “unitAll LTCRH in the Health Departments were studied, less a center located in an institution that included an NH and was included in our study of COVID-19 in NH” the statement requires clarification.

-We are clarified this phrase.

Results

Replace Si by yes in the table 1. We correct the mistake.

Line 152 “A with fewer than 30 places in 6 centers,” (With fewer than 30 places….). We correct the mistake.

Lines 219-221 “In addition, the means of ratios for residents/staff and nursing assis-219 tants/residents in the LTCRH were according to the legislation of Valencia Generali-220 tat[10]. However, the ratio of registered nurses/residents was low, and two centers had 221 more than 40 residents, the maximum size allowed for these LTCRH[10]”.

Would you clarify the relationship between this statement and the above mentioned results!

-We compare the COVID-19 incidence in nursing homes with COVID-19 incidence in long-term care residential homes in Castellon from our study:

Arnedo-Pena, A.; Romeu-Garcia, M.A.; Gascó-Laborda, J.C.; Meseguer-Ferrer, N.; Safont-Adsuara, L.; Prades-Vila, L.; Flores-Medina, M.; Rusen, V.; Tirado-Balaguer, M.D.; Sabater-Vidal, S.; et al. Incidence, Mortality, and Risk Factors of COVID-19 in Nursing Homes. Epidemiologia 2022,3,179-190.

Line 236 “results of Gorgels y co-authors [31]” (results of Gorgels and co-authors [31]). We correct the mistake.

Reviewer 2 Report

The submitted manuscript provides very interesting results dealing with incidence, hospitalization, mortality and risk factors of COVID-19 in long-term care residential homes for patients with chronic mental illness. This topic is very current and so far relatively understudied. The introduction is comprehensible and legibly written.

In the introduction, I lack information about the symptoms and the course of the COVID-19 disease. The authors should more characterize this disease.

I have the following comments on the text (mainly about correcting spaces):

line 2- “ … 2Public Health …” instead of “ … 2 Public Health…”

line 43- “ … elevated[1-3] …” instead of “ … elevated [1-3]…”

line 45- “ … are living[4-6]…” instead of “ … are living [4-6]…”

line 47- “ … established[7-9]. …” instead of “ … established [7-9]…”

line 53- “ …  hospitalization[10]. …” instead of “ …  hospitalization [1].…”

line 65- “ …studies[12]. …” instead of “ …  studies [12].…”

line 67- “ …Public Health     Center of Castellon …” instead of “ …  Public Health Center of Castellon…”

line 71- “ …SARS-CoV-2[Roche-TIB …” instead of “ …SARS-CoV-2 [Roche-TIB…”

line 74-75- “ … (Biomé- 74 rieux, F-69280, Marcy l’Etoile, France)…” instead of “ …[Biomé- 74 rieux, F-69280, Marcy l’Etoile, France]…”

line 75-76 "... (PANBIO™ 75 COVID-19 Ag, Abbott Laboratory, Abbott Park, IL, USA)...” instead of “ …[PANBIO™ 75 COVID-19 Ag, Abbott Laboratory, Abbott Park, IL, USA]…”

line 78- “ …(Alinity,Abbott Laboratory, Abbott Park, IL, USA)…” instead of “ …[Alinity,Abbott Laboratory, Abbott Park, IL, USA]…”

line 79- “ …cases[13-15]…” instead of “ …cases [13-15]…”

line 83-90- “ …In this questionnaire, different variables were collected:

Numbers of residents, mean age, numbers of women and men, disability grade, …” instead of “ …In this questionnaire, different variables were collected: numbers of residents, mean age, numbers of women and men, disability grade,…”

line 104- “ …-Residents:…” instead of “ …- Residents:…”

line 109- “ …-Staff:…” instead of “ …- Staff:…”

line 112- “ …-LTCRH:…” instead of “ …- LTCRH:…”

line 114- “ … area        (≥10,000–49,999 inhabitants),…” instead of “ … area (≥10,000–49,999 inhabitants),…”

line 127- “ …and        multilevel Poisson…” instead of “ …and multilevel Poisson…”

line 133- “ …was    adjusted    for…” instead of “ …was adjusted for…”

line 135- “ … (Stata Corp, College Station,TX, USA)…” instead of “ … [Stata Corp, College Station,TX, USA]…”

line 148- “ …illness      (II).Health…” instead of “ …illness (II). Health…”

Table 1- Residents N=346” should be justified

line 178- “ …0.01%-1.50%).Two…” instead of “ …0.01%-1.50%). Two…”

line 179- “ .... and      were two…” instead of “ …and were two…”

Table 2- ,,Staff n=482" instead ,,Staff n=482"

line 193-195 “ ....Adjusted.6” instead of “ …Adjusted. 6…” and the same  situation with the numbers: 7,8,9,10,13

line 212-213- “ ....5Adjusted.6” instead of “ …5 Adjusted. 6…” and the same situation whith the numbers: 8,9.

line 219- “ .... versus 0.6%[11]…” instead of “ …versus 0.6% [11].…”

line 220- “ .... Generalitat[10].…” instead of “ …Generalitat [10].…”

line 222- “ .... LTCRH[10] .…” instead of “ …LTCRH [10].…”

line 224- “ .... hospitalization[21-23].…” instead of “ …hospitalization [21-23]…”

line 234- “ .... Canada[30]..…” instead of “ …Canada [30].…”

line 240- “ .... NHs[33-35]...…” instead of “ …NHs [33-35].…”

line 247- “ .... incidence[38]. ...…” instead of “ … incidence [38].…”

line 265- “ .... priority[39-41] ...…” instead of “ …priority [39-41]…”

Author Response

  1. Second Reviewer.

The submitted manuscript provides very interesting results dealing with incidence, hospitalization, mortality and risk factors of COVID-19 in long-term care residential homes for patients with chronic mental illness. This topic is very current and so far relatively understudied. The introduction is comprehensible and legibly written.

-We appreciate your revision and your comment.

In the introduction, I lack information about the symptoms and the course of the COVID-19 disease. The authors should more characterize this disease.

Thank you very much for your useful indication. We mention the epidemiologic characteristics of the disease and the transmission in long-term care residential homes.

I have the following comments on the text (mainly about correcting spaces):

We have corrected all the indications.

line 2- “ … 2Public Health …” instead of “ … 2 Public Health…”

line 43- “ … elevated[1-3] …” instead of “ … elevated [1-3]…”

line 45- “ … are living[4-6]…” instead of “ … are living [4-6]…”

line 47- “ … established[7-9]. …” instead of “ … established [7-9]…”

line 53- “ …  hospitalization[10]. …” instead of “ …  hospitalization [1].…”

line 65- “ …studies[12]. …” instead of “ …  studies [12].…”

line 67- “ …Public Health     Center of Castellon …” instead of “ …  Public Health Center of Castellon…”

line 71- “ …SARS-CoV-2[Roche-TIB …” instead of “ …SARS-CoV-2 [Roche-TIB…”

line 74-75- “ … (Biomé- 74 rieux, F-69280, Marcy l’Etoile, France)…” instead of “ …[Biomé- 74 rieux, F-69280, Marcy l’Etoile, France]…”

line 75-76 "... (PANBIO™ 75 COVID-19 Ag, Abbott Laboratory, Abbott Park, IL, USA)...” instead of “ …[PANBIO™ 75 COVID-19 Ag, Abbott Laboratory, Abbott Park, IL, USA]…”

line 78- “ …(Alinity,Abbott Laboratory, Abbott Park, IL, USA)…” instead of “ …[Alinity,Abbott Laboratory, Abbott Park, IL, USA]…”

line 79- “ …cases[13-15]…” instead of “ …cases [13-15]…”

line 83-90- “ …In this questionnaire, different variables were collected:

Numbers of residents, mean age, numbers of women and men, disability grade, …” instead of “ …In this questionnaire, different variables were collected: numbers of residents, mean age, numbers of women and men, disability grade,…”

line 104- “ …-Residents:…” instead of “ …- Residents:…”

line 109- “ …-Staff:…” instead of “ …- Staff:…”

line 112- “ …-LTCRH:…” instead of “ …- LTCRH:…”

line 114- “ … area        (≥10,000–49,999 inhabitants),…” instead of “ … area (≥10,000–49,999 inhabitants),…”

line 127- “ …and        multilevel Poisson…” instead of “ …and multilevel Poisson…”

line 133- “ …was    adjusted    for…” instead of “ …was adjusted for…”

line 135- “ … (Stata Corp, College Station,TX, USA)…” instead of “ … [Stata Corp, College Station,TX, USA]…”

line 148- “ …illness      (II).Health…” instead of “ …illness (II). Health…”

Table 1- “ Residents N=346” should be justified

line 178- “ …0.01%-1.50%).Two…” instead of “ …0.01%-1.50%). Two…”

line 179- “ .... and      were two…” instead of “ …and were two…”

Table 2- ,,Staff n=482" instead ,,Staff n=482"

line 193-195 “ ....Adjusted.6…” instead of “ …Adjusted. 6…” and the same  situation with the numbers: 7,8,9,10,13

line 212-213- “ ....5Adjusted.6…” instead of “ …Adjusted. 6…” and the same situation whith the numbers: 8,9.

line 219- “ .... versus 0.6%[11]…” instead of “ …versus 0.6% [11].…”

line 220- “ .... Generalitat[10].…” instead of “ …Generalitat [10].…”

line 222- “ .... LTCRH[10] .…” instead of “ …LTCRH [10].…”

line 224- “ .... hospitalization[21-23].…” instead of “ …hospitalization [21-23]…”

line 234- “ .... Canada[30]..…” instead of “ …Canada [30].…”

line 240- “ .... NHs[33-35]...…” instead of “ …NHs [33-35].…”

line 247- “ .... incidence[38]. ...…” instead of “ … incidence [38].…”

line 265- “ .... priority[39-41] ...…” instead of “ …priority [39-41]…”